# Prevalence of metabolic syndrome among Iranian postmenopausal females: A systematic review and meta-analysis

Erfan Ghadirzadeh[1,2], Atoosa Mahmoodi[3], Naseh Yousefi[4], Pedram Nezhadnaderi[5], Mojgan Geran[6,7], Morteza Biabani[1], Anita Ziari[1], Mobina Gheibi[8,9], Mahmood Moosazadeh[2]*, Maryam Zarrinkamar[6,7]*

1 Student Research Committee, School of Medicine, Mazandaran University of Medical Sciences, Sari, Iran, 2 Gastrointestinal Cancer Research Center, Non-Communicable Disease Institute, Mazandaran University of Medical Sciences, Sari, Iran, 3 Islamic Azad University, Sanandaj Branch, School of Nursing, Sanandaj, Iran, 4 Neuromusculoskeletal Research Center, Department of Physical Medicine and Rehabilitation, School of Medicine, Iran University of Medical Sciences, Tehran, Iran, 5 Student Research Committee, School of Medicine, Shiraz University of Medical Sciences, Shiraz, Iran, 6 Department of Family Medicine, Faculty of Medicine, Mazandaran University of Medical Sciences, Sari, Iran, 7 Diabetes Research Center, Institute of Herbal Medicines and Metabolic Disorders, Mazandaran University of Medical Sciences, Sari, Iran, 8 Student Research Committee, School of Allied Medical Sciences, Mazandaran University of Medical Sciences, Sari, Iran, 9 Department of Medical Laboratory Sciences, Razi Hospital, Mazandaran University of Medical Sciences, Qaemshahr, Iran

* mmoosazadeh1351@gmail.com (MM); Maryamzarrinkamar3@gmail.com (MZ)

## Abstract

### Introduction

This systematic review and meta-analysis aimed to estimate the prevalence of metabolic syndrome (MetS) among Iranian postmenopausal women by addressing inconsistencies in prior research and providing reliable data to inform evidence-based policies for reducing Iran's MetS burden.

### Methods

Medline/PubMed, Scopus, Embase, Web of Science, Google Scholar, IMEMR, SID, MagIran, ISC, IranDoc/Ganj, Civilica, and RPIS were searched from their dates of inception until April 2025. The quality of the evidence was assessed using the Joanna Briggs Institute critical appraisal checklist. The prevalence of MetS was calculated using the random effects model using Stata version 17. Additionally, sensitivity analysis, subgroup analysis, meta-regression, and publication bias were assessed. The protocol is registered in PROSPERO, number CRD420251039469.

### Results

A total of 24 papers were enrolled, comprising 17,281 postmenopausal participants with a pooled estimate of 58.42% (95%CI: 52.35–64.48, $I^2$: 98.59%, Q: 836.97) MetS among Iranian postmenopausal females. The prevalence of MetS was 64.10%,

**Data availability statement:** All relevant data are within the paper and its Supporting information files.

**Funding:** The author(s) received no specific funding for this work.

**Competing interests:** None.

47.01%, 63.24%, and 50.16% in high-quality, medium-quality, population-based, and institutional-based studies, respectively. Moreover, meta-regression and subgroup analyses demonstrated study quality, study setting, and age as considerable sources of heterogeneity.

## Conclusion

This study highlights a high prevalence of MetS (≈58.5%) among Iranian postmenopausal women, with even greater estimates in high-quality and population-based studies, which underscores a significant public health concern. Given this substantial burden, routine screening for MetS components should be integrated into standard care for postmenopausal women, complemented by public health initiatives targeting lifestyle modifications and broader preventive strategies.

## Introduction

Metabolic syndrome (MetS), characterized by abdominal obesity, hypertension, dyslipidemia, and hyperglycemia, is a significant predictor of cardiovascular disease (CVD), type 2 diabetes mellitus (T2DM), stroke, and premature mortality [1]. In Iran, a 2025 meta-analysis demonstrated that MetS affects approximately 30% of adults, contributing substantially to the national burden of non-communicable diseases (NCDs) [2]. The co-occurrence of these metabolic abnormalities accelerates atherosclerosis, exacerbates systemic inflammation, and amplifies oxidative stress, thereby heightening the risk of chronic morbidity and mortality [3,4].

Menopause, characterized by a significant drop in estrogen, directly increases the risk of developing MetS through several interconnected pathways [5–7]. Estrogen loss promotes the buildup of visceral fat, which drives insulin resistance and inflammation, up-regulates hepatic insulin clearance, predisposing to hyperglycemia, decreases lipoprotein lipase activity, thereby inducing dyslipidemia, and impairs endothelial function by reducing nitric oxide synthase-mediated vasodilation, culminating in elevated blood pressure [8–11].

Consequently, postmenopausal women exhibit a disproportionately higher prevalence of MetS compared to their male and premenopausal counterparts; with estimates ranging from 28% to 88% across various Iranian populations [12,13]. These biological vulnerabilities, compounded by age-related and lifestyle factors, underscore the need for focused epidemiological assessments within this demographic. Nevertheless, while a comprehensive global systematic review reported a pooled MetS prevalence of approximately 37% among postmenopausal women worldwide [14], the rates observed in Iranian studies were notably higher. This discrepancy underscores that the burden in Iran is particularly severe and highlights the need for a precise, updated national estimate to inform local policymakers.

From a public health perspective, postmenopausal women represent a critical demographic for targeted MetS screening and management. Despite their elevated risk, this group often falls through the cracks of healthcare systems. In Iran, as in

many settings, NCD services can be fragmented; gynecological care typically focuses on reproductive health, while management of MetS components like hypertension and dyslipidemia falls under the purview of primary care or internal medicine. This siloed approach, combined with the often asymptomatic nature of early MetS, may lead to under-diagnosis and a missed opportunity for preventive interventions [15,16].

An updated and robust prevalence estimate is needed to inform and refine several key national health policies in Iran. These include the Iran Package of Essential Non-communicable (IraPEN) Diseases Interventions, which aims to integrate NCD prevention and control into primary healthcare. Our findings can provide the evidence base to advocate for the inclusion of routine MetS screening within the standard care protocol for middle-aged and postmenopausal women attending primary health centers, ensuring that the substantial burden borne by postmenopausal women is adequately reflected in national targets, resource allocation, and the design of public awareness campaigns [17,18].

While previous research has explored the prevalence of MetS in Iranian postmenopausal women, existing reviews suffer from several limitations for current use. Notably, since the publication of these earlier reviews in 2018 by Ebtekar et al. [19] and Tabatabaei-Malazy et al. [20], a significant body of new evidence has emerged, including data from several recent population-based cohorts across Iran (e.g., the Tabari cohort, Hoveyzeh cohort, Fasa cohort, Bushehr Elderly Health Program, and Birjand Longitudinal Aging Study, among others) which capture regionally diverse populations [21–26]. These population-based cohorts provide robust, recent data essential for refining prevalence estimates.

This expanding evidence base, coupled with the methodological limitations of past syntheses, including a limited search strategy, inconsistent diagnostic criteria which leads to methodological heterogeneity, omission of several relevant studies, absence of sensitivity analyses, and failure to investigate heterogeneity sources using established methods such as subgroup analyses or meta-regression, reduce the reliability and applicability of their findings and necessitates an updated and more comprehensive systematic review and meta-analysis.

Considering the wide range of reported prevalence rates in the primary studies and the shortcomings of previous reviews, incorporating high-quality evidence, particularly from the most recent cohorts, will enhance the precision and generalizability of findings, enabling evidence-based policy updates to mitigate Iran's MetS burden. Therefore, this systematic review and meta-analysis aimed to provide an updated and comprehensive estimate of the prevalence of MetS among Iranian postmenopausal women.

## Methods

This systematic review and meta-analysis aimed to assess the prevalence of MetS in Iranian postmenopausal females and it was reported according to Preferred Reporting Items for Systematic Reviews and Meta-Analyses (PRISMA) guidelines [27], with its protocol being registered on the PROSPERO database (CRD420251039469).

### Eligibility criteria

Studies reporting the prevalence of MetS (or all of its five components) in Iranian postmenopausal females were included in this systematic review based on the following POLIS (Participants, Outcome, Location, Indicator, and Study Design) criteria without any language restrictions.

*Participants:* Any study that included Iranian postmenopausal (defined as cessation of menstrual period for at least 12 months) females either as a whole or as a part of the studied population was included. This definition based on amenorrhea, rather than a predetermined age cutoff, was employed to accurately capture the postmenopausal status across all included studies, accounting for potential variations in the natural age at menopause within the Iranian population. Exclusion criteria were: 1. Studies conducted only on postmenopausal females with MetS (due to inability to extract prevalence data), 2. Studies that were conducted on surgically-induced menopause patients, 3. Studies with less than 30 sample size, and 4. Multiple reports from the same parent study or cohort, or studies with duplicate (and/or shared and/or overlapping) participant populations (only the most complete or recent report was included).

*Outcome:* studies that reported the prevalence of MetS (or its components) among postmenopausal females.

*Location:* Iran.

*Indicator:* prevalence of MetS.

*Study design:* observational studies and clinical trials conducted on Iranian postmenopausal females with relevant data on MetS (or its components) at baseline.

Additionally, case reports, case series, animal studies, and articles with unavailable full-texts were excluded from the study.

## Search strategy

A systematic search was performed in international (including Medline/PubMed, Google Scholar, Scopus, Web of Science, Embase, and The Index Medicus for the Eastern Mediterranean Region) and local databases/search engines (including Scientific Information Database, MagIran, Islamic World Science Citation Center, IranDoc/Ganj, Civilica, and Research Proposal Information System), from their dates of inception until April 2025 with no language limitation. Furthermore, we conducted a manual review of gray literature and reference lists of included papers to ensure the inclusion of relevant studies that may have been missed in electronic searches.

Two researchers (AZ and MB) utilized keywords including metabolic syndrome, Reaven syndrome x, insulin resistance syndrome, metabolic cardiovascular syndrome, dysmetabolic syndrome, cardiometabolic syndrome, metabolic dysfunction syndrome, metabolic disorder syndrome, deadly quartet, cardiometabolic impairment, menopause, postmenopausal, permanent cessation of menstruation, and Iran, along with their combinations and related synonyms based on Medical Subject Headings (MeSH) and free-text methods, to identify relevant literature. The search syntax is available in S1 File.

## Screening

Two assessors (MG and AM) first independently evaluated the titles and abstracts of all retrieved records to pinpoint relevant research, removing duplicates and unrelated records. They subsequently reviewed the full-texts of potential studies separately to determine the final study selections. At this stage, any papers that had collected relevant information regarding the key variables of interest (menopausal status and MetS diagnosis) but did not explicitly report the primary outcome of interest (prevalence of MetS among their postmenopausal population) were identified. The corresponding authors of these studies were subsequently contacted via email to request additional data on this outcome, if available. Any disagreements were resolved by an additional third reviewer (MZ). The reference lists of previous systematic reviews on this topic were also cross-checked to ensure all relevant primary studies were considered for eligibility.

## Data extraction

Two reviewers (NY and PN) independently extracted data regarding the following variables from included papers into a Microsoft Excel spreadsheet with disagreements being resolved by a third reviewer (EG):

- Study characteristics (Author, country, publication year, Study location/province, population, study setting [population-based or institutional], and sample size).

- Study design (e.g., case-control, cohort, cross-sectional).

- Population characteristics (age and age at menopause).

- MetS diagnostic criteria. To ensure methodological consistency across studies, when multiple diagnostic criteria for MetS were reported, the NCEP ATP III definition was preferentially selected as the reference standard due to its widespread adoption in clinical research in Iran. This approach was pursued to minimize methodological heterogeneity in case ascertainment between the included studies. While we acknowledge that other criteria (e.g., IDF) incorporate

population-specific waist circumference cut-offs, the consistent application of NCEP ATP III across the majority of Iranian studies was deemed the most appropriate approach to ensure comparability and reduce a significant source of variation in our pooled estimate.

- Outcome (proportion of postmenopausal females diagnosed with MetS).

Furthermore, studies with duplicate or overlapping study populations were identified during this phase. Only the study encompassing the larger or more comprehensive population was enrolled in the review to avoid redundancy.
For the purpose of this review, study settings were categorized as follows:

- Population-Based Studies: These are studies that recruit their participants from a defined general population. The sampling frame is designed to be representative of the community at large, and participants are not selected based on their attendance at a specific healthcare facility. Examples include large cohort studies (e.g., the Tabari Cohort, Tehran Lipid and Glucose Study) and cross-sectional surveys that randomly sample households or individuals from a city or region.

- Institution-Based Studies: These are studies that recruit their participants from a specific institution or clinical setting. The sampling frame is confined to individuals who are present in that institution, typically for a specific reason, which introduces a potential for selection bias. Examples include studies recruiting patients from menopause clinics, diabetes clinics, hospitals, bone densitometry units, or groups of teachers or employees from a specific organization.

## Quality assessment

To evaluate possible bias in the selected studies, two reviewers (EG and MG) critically appraised included papers using the Joanna Briggs Institute (JBI) critical appraisal checklist for studies reporting prevalence data [28], separately, and disagreements were resolved by an expert third reviewer (MM). The JBI checklist for prevalence studies is a structured method to examine the methodological rigor of research included in systematic reviews. It comprises nine key criteria, each rated as "yes," "no," "unclear," or "not applicable" based on the potential risk of bias. Each "Yes" answer equals one score. The sum of scores from the nine questions will eventually determine the overall quality of the paper based on the following scoring system: 0–3: Low quality, 4–6: Medium quality, and 7–9: High quality.

## Statistics

The statistical procedures were conducted utilizing Stata software, version 17 (StataCorp LLC). A random-effects model with restricted maximum likelihood (REML) was implemented to calculate the pooled prevalence of MetS within post-menopausal populations. Effect sizes (prevalence) derived from individual primary studies and the overall estimate were graphically depicted in a forest plot, along with 95% confidence intervals (CI). Heterogeneity was quantified through the application of the $I^2$ index and Cochran's Q heterogeneity test and visualized by Galbraith plot [29]. Potential publication bias was evaluated via visual inspection of the Doi plot [30,31] and its corresponding LFK index, complemented by Egger's and Begg's tests. The trim-and-fill method was further employed to adjust for potential missing studies.
To investigate potential sources of heterogeneity, meta-regression techniques along with subgroup analysis were applied, incorporating covariates such as study quality, study setting, study region (categorized according to Iranian Ministry of Interior classification in June 2014 based on factors such as provincial adjacency, geographical proximity, and shared regional characteristics, S2 File), chronological age, menopausal age at onset, study year (categorized as studies on and before 2010, 2011–2015, 2016–2020, and after 2020), study sample size frame (categorized as follows: studies with less than 200 sample size were categorized as "very small", with 200–400 sample size were categorized as "small", with 400–1000 were categorized as "medium", and with more than 1000 were categorized as "large"), healthcare price index category, and primary care center density category.

*Statistics Center of Iran* (https://amar.org.ir/) ranked each Iranian province based on by the price index of consumer goods and services for households in the healthcare sector (healthcare price index), and the density of primary health care centers per 100,000 population (primary care center density) (https://amar.org.ir/development-indicators#5029327---). The rankings ranged from 1 to 31 (covering all 31 provinces in Iran). Using the year 2023 rankings, the provinces were divided into three levels in each index: high (ranks 1–10), medium (ranks 11–20), and low (ranks 21–31). As the price of health services and the availability of primary care centers were thought to be possibly effective on MetS diagnosis, these indicators were used for further subgroup analysis of the included studies for heterogeneity assessment purposes. The robustness of the synthesized estimate was examined through leave-one-out sensitivity analysis, systematically excluding individual studies to assess their influence on the overall estimate.

### Ethics

This systematic review and meta-analysis involved synthesis of previously published data, no human or animal subjects were recruited, and no primary original data was collected; thus, no institutional review board approval was applicable.

## Results

### Study selection process

The study selection process, outlined in Fig 1, commenced with 2,559 articles identified through searches. Following the implementation of a systematic search strategy, 641 duplicate entries were eliminated. Subsequent screening phases involved title and abstract reviews, resulting in the exclusion of 1,698 articles deemed irrelevant to the research focus, alongside 5 publications with inaccessible full-text content. From the remaining 215 full-text articles subjected to eligibility evaluation, 17 were excluded due to geographical restrictions (non-Iranian populations), 5 for insufficient sample sizes (n<30), and 12 for failing to include postmenopausal women within their study cohorts (or menopausal status unknown). Methodological exclusions comprised 18 investigations limited exclusively to postmenopausal populations with confirmed MetS (precluding prevalence calculations), 24 studies lacking explicit prevalence data (despite attempted correspondence with authors), and 1 study conducted solely on women with surgically-induced menopause [32]. An additional 107 articles were excluded for thematic irrelevance, while 12 publications were removed due to population overlap, with priority given to studies encompassing larger or more representative samples [33–45]. Ultimately, 24 observational investigations comprising 22 cross-sectional analyses and 2 prospective cohort studies satisfied all eligibility criteria for quantitative synthesis.

### Study characteristics

The included studies were conducted across various provinces, with 7 papers in Tehran, 5 papers in Mazandaran, and 12 papers in other provinces comprising 17,281 postmenopausal participants. Sample sizes ranged from 72 to 2653 postmenopausal participants. Nine studies were conducted in an institutional setting, while 15 studies were conducted with a population-based design. Also, MetS diagnosis was established using NCEP ATP III criteria in all studies. The characteristics of the included studies are shown in Table 1.

### Quality assessment

Quality assessment was performed using JBI quality assessment tool with scores ranging from 5 to 9, including 8 studies with moderate quality and 16 studies with high quality. Details of quality assessment can be found on S3 File.

### Prevalence of MetS among Iranian postmenopausal females

A total of 24 papers were enrolled to assess the prevalence of MetS among postmenopausal females, with the lowest prevalence rate in Abbasi et al.'s study (28%) in Qazvin [12] and the highest rate in Nakhjavani et al.'s study (87.5%) in

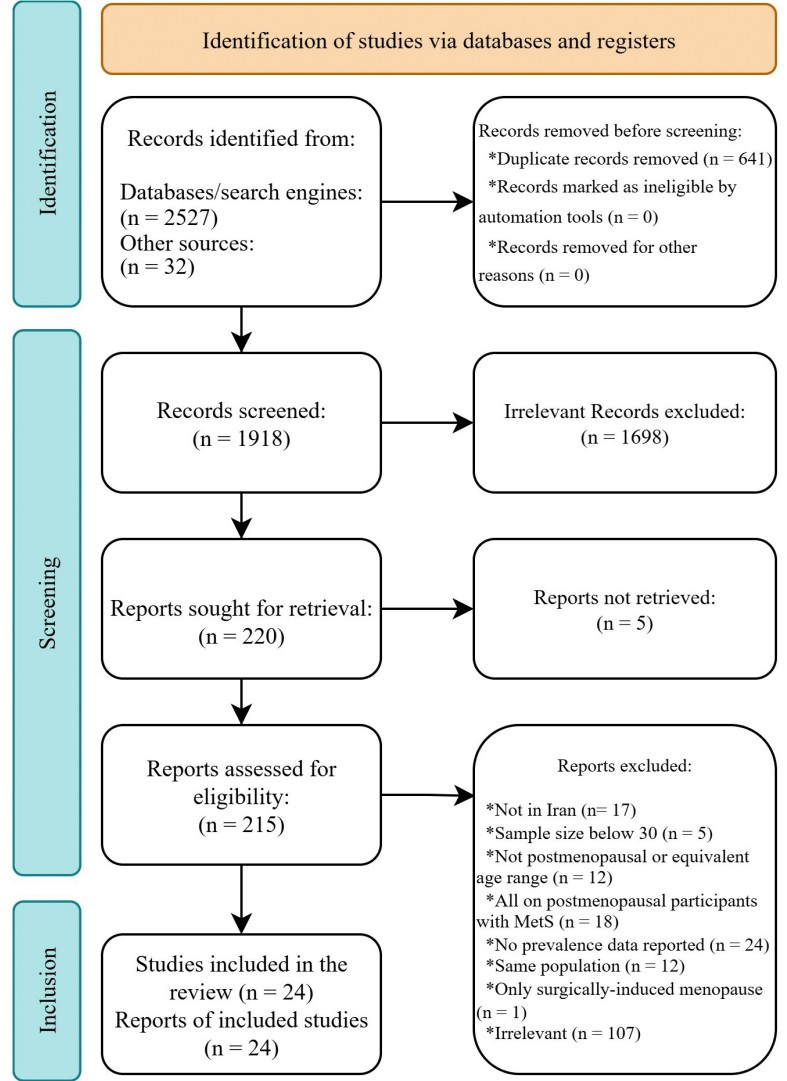

**Fig 1. PRISMA flowchart of screening and selection process of qualified studies.**

Tehran [13]. Results indicated a pooled prevalence of 58.42% (95%CI: 52.35–64.48, I²: 98.59%, Q: 836.97) MetS among Iranian postmenopausal females with considerable levels of statistical heterogeneity (Fig 2). Sensitivity analysis showed no significant change in the overall prevalence after the exclusion of studies one at a time (Fig 3). Additionally, the results of Egger's (β: −6.99, P < 0.001) and Begg's (Kendall's τ: −94, P: 0.021) tests along with Doi plot (Fig 4) showed considerable publication bias (LFK index: −2.64 indicating asymmetry); however, the trim and fill analysis showed no additional publications.

## Sources of heterogeneity (Subgroups analysis and meta-regression)

Table 2 shows the results of subgroup analysis, indicating a significant difference in the prevalence of MetS based on the quality of included studies and the setting in which included studies were conducted (See S4 File for forest plots of subgroup analyses). The pooled prevalence of MetS among Iranian postmenopausal females was 64.10%

**Table 1. Characteristics of included primary studies.**

| Study | Year | City | Design | Setting | Population | Dx | Chronological Age (year) | Menopausal Age (year) | Sample Size (number) | | Quality |
|---|---|---|---|---|---|---|---|---|---|---|---|
| | | | | | | | | | PM | PM+MetS | |
| Abbasi [12] | 2017 | Qazvin | CS | I | PM women referred for BD | ATP III | NR | NR | 143 | 40 | M |
| Ayni [75] | 2007 | Tehran | CS | PB | TLGS Baseline | ATP III | 59.5±5.2 | NR | 1645 | 1088 | H |
| Heidari [76] | 2015 | Amirkola | CS | PB | AHAP | ATP III | 67.9±6.7 | NR | 537 | 449 | H |
| Nabipour [77] | 2010 | Bushehr | CS | PB | IMOS | ATP III | 58.6±7.4 | NR | 382 | 261 | H |
| Delavar [78] | 2009 | Babol | CS | PB | Women from 14 urban PHC's registered households | ATP III | NR | NR | 135 | 52 | H |
| Ebrahimpour [79] | 2010 | Tehran | CS | PB | Adult residents of the 17th district of Tehran | ATP III | 56.5±5.8 | NR | 285 | 171 | H |
| Eshtiaghi [80] | 2010 | Tehran | CS | PB | Non-institulized women | ATP III | 57.0±7.3 | 47.8±5.4 | 215 | 115 | H |
| Farahmand [81] | 2017 | Tehran | CH | PB | TLGS Follow-Up | ATP III | 55.1±4.3 | NR | 408 | 283 | H |
| Soleimani [82] | 2018 | Ishafahn, Najafabad, Arak | CH | PB | IHHP | ATP III | 60.0±93.1 | 48.1±5.5 | 1154 | 674 | H |
| Namazi Shabe-stari [83] | 2016 | Tehran | CS | I | Menopause clinic of Tehran women's general hospital | ATP III | 53.9±5.5 | 47.2±5.2 | 264 | 108 | M |
| Maharlouei [84] | 2013 | Shiraz | CS | PB | SWHCS | ATP III | 58.6±6.7 | NR | 434 | 222 | M |
| Marjani [85] | 2012 | Gorgan | CS | I | Women referred to the health centers in Gorgan | ATP III | 54.1±5.2 | NR | 100 | 31 | M |
| Montazeri [22] | 2023 | Sari | CS | PB | TCS | ATP III | NR | NR | 2653 | 1672 | H |
| Moradi [25] | 2024 | Hoveyzeh | CS | PB | HCS | ATP III | 58.0±6.1 | NR | 1313 | 796 | H |
| Naghipour [86] | 2022 | Somesara | CS | PB | GCS | ATP III | NR | 47.54±5.7 | 2377 | 1582 | H |
| Nakhjavani [13] | 2014 | Tehran | CS | I | Women from diabetes clinic of Vali-Asr hospital | ATP III | 60.3±0.3 | NR | 418 | 366 | M |
| Sayahi [87] | 2015 | Ahvaz | CS | I | PM women referred to health centers in Ahvaz | ATP III | 55.4±6.0 | 49.0±3.6 | 165 | 107 | H |
| Shahvazi [88] | 2016 | Yazd | CS | I | Female teachers from 84 schools | ATP III | NR | NR | 72 | 45 | H |
| Zareei [26] | 2022 | Fasa | CS | PB | FCS | ATP III | NR | NR | 2165 | 1305 | H |
| Bakhtiari [89] | 2018 | Babol | CS | I | Women in rural health clinics of the central part of Babol | ATP III | 64.2±3.8 | NR | 164 | 75 | M |
| Rabiei [24] | 2021 | Bushehr | CS | PB | BEHP2 | ATP III | 69.1±6.3 | NR | 1256 | 849 | H |
| Sadat [90] | 2015 | Tehran | CS | I | Nursing homes of Tehran | ATP III | NR | NR | 155 | 73 | M |
| Saeedi [23] | 2023 | Birjand | CS | PB | BLAS | ATP III | NR | NR | 701 | 543 | H |
| Ziaei [91] | 2011 | Ahvaz | CS | I | Non-Surgical PM women in Ahvaz | ATP III | 56.7±6.0 | 49.5±3.3 | 140 | 60 | M |

**Dx:** Diagnostic criteria of MetS, **PM:** Postmenopausal, **MetS:** Metabolic Syndrome, **M:** Moderate, **H:** High, **BD:** Bone Densitometry, **NR:** Not Reported, **CS:** Cross-Sectional, **CH:** Cohort (Longitudinal study), **I:** Institutional, **PB:** Population-Based, **ATP III:** National Cholesterol Education Program-Adult Treatment Panel III (NCEP ATP III), **TLGS:** Tehran Lipid & Glucose Study, **AHAP:** Amirkola Health and Ageing Project, **IMOS:** Iranian Multicentral Osteoporosis Study, **IHHP:** Isfahan Healthy Heart Program, **SWHCS:** Shiraz Women's Health Cohort Study (Imam Khomeini Relief Foundation eligible households), **TCS:** Tabari Cohort Study, **HCS:** Hoveyzeh Cohort Study, **GCS:** Guilan Cohort Study, **FCS:** Fasa Cohort Study, **BEHP2:** Bushehr Elderly Health Program 2nd stage, **BLAS:** Birjand Longitudinal Aging Study, **PHC:** Public Health Centers.

(95%CI: 59.40–68.81) in high-quality papers and 47.01% (95%CI: 34.12–59.89) in medium-quality papers, indicating this variable as a source of heterogeneity. Also, population-based studies demonstrated a higher prevalence rate (63.24%, 95%CI: 58.00–68.49) compared to studies conducted in an institutional setting (50.16%, 95%CI: 37.76–62.57) which

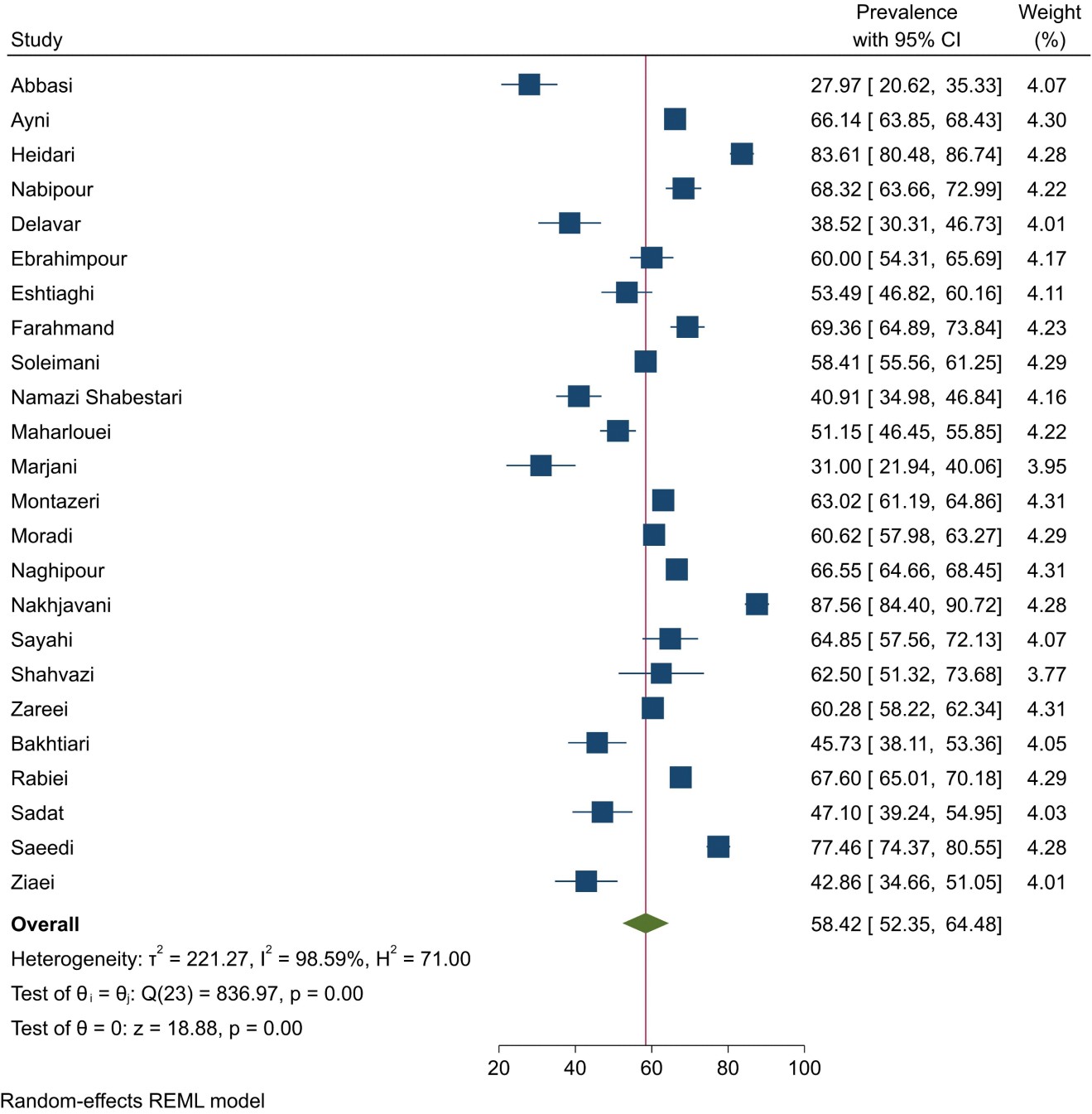

**Fig 2. Forest plot demonstrating prevalence of MetS among Iranian postmenopausal females in both individual primary studies and the overall estimate with 95%CI.**

may indicate study setting as a possible source of heterogeneity. However, when subgroup analysis was done based on regional subgroups, sample size frames, healthcare price index, and primary care center density, no considerable changes were observed in the overall estimate or the $I^2$ index. Also, subgroup analysis based on study year did not yield any observable trends.

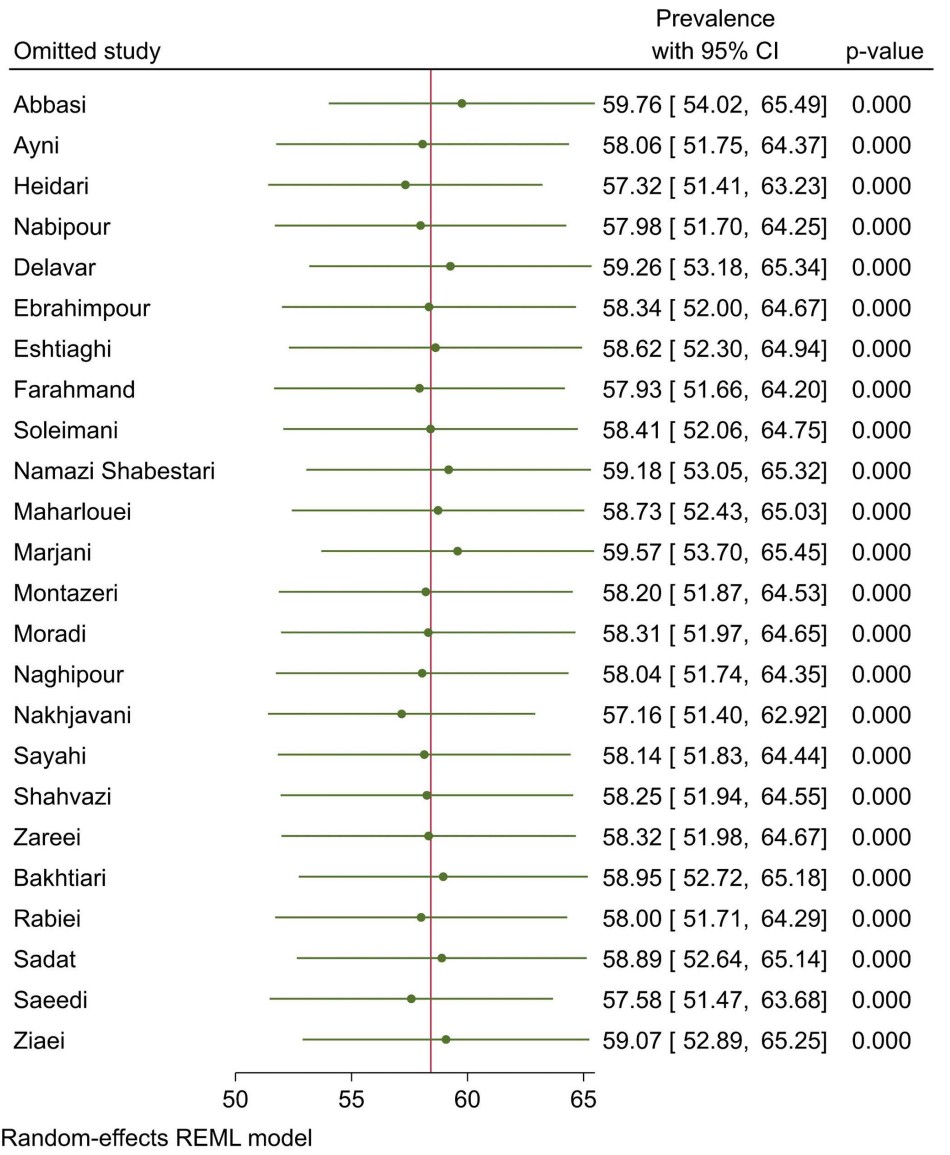

**Fig 3. Sensitivity analysis by leave-one-out method indicating none of the primary studies had a significant influence on the overall estimate.**

In order to further assess the sources of heterogeneity among participants, a meta-regression analysis was conducted, demonstrating study quality (β: 7.13, SE: 2.03, z: 3.51, $R^2$: 33.66, P < 0.001), study setting (β: 12.86, SE: 5.98, z: 2.15, $R^2$: 13.95, P: 0.032), age (β: 1.54, SE: 0.74, z: 2.07, $R^2$: 17.84, P: 0.038), and sample size frame (β: 7.01, SE: 2.12, z: 3.30, $R^2$: 31.00, P: 0.001) as considerable sources of heterogeneity. However, results of meta-regression for study region (β: 2.41, SE: 2.30, z: 1.05, $R^2$: 0.24, P: 0.294), participants age at menopause (β: −0.77, SE: 5.79, z: −0.13, $R^2$: 0.00, P: 0.894), study year (β: 1.85, SE: 2.89, z: 0.64, $R^2$: 0.00, P: 0.521), healthcare price index (β: −2.11, SE: 3.73, z: −0.57, $R^2$: 0.00, P: 0.571), and primary care center density (β: 1.23, SE: 3.86, z: 0.32, $R^2$: 0.00, P: 0.750) was insignificant.

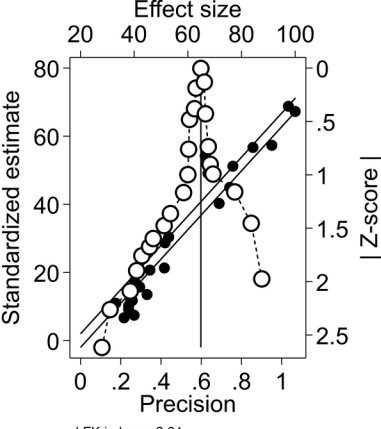

LFK index = -2.64

**Fig 4. Doi and Galbraith plots indicating significant publication bias (asymmetry and a LFK index of −2.64) and considerable level of heterogeneity.**

**Table 2. Subgroup analysis of the prevalence of MetS among Iranian postmenopausal females.**

| Subgroups | | Number of Studies | Pooled Prevalence | 95%CI | Statistical Heterogeneity | | |
|---|---|---|---|---|---|---|---|
| | | | | | I² (%) | Q | P |
| Quality | High | 16 | 64.10 | 59.40-68.81 | 97.28 | 320.65 | < 0.001 |
| | Moderate | 8 | 47.01 | 34.12-59.89 | 97.29 | 497.24 | < 0.001 |
| Setting | Population Based | 15 | 63.24 | 58.00-68.49 | 97.90 | 354.63 | < 0.001 |
| | Institutional | 9 | 50.16 | 37.76-62.57 | 96.77 | 478.59 | < 0.001 |
| Region* | 1 | 13 | 55.24 | 45.01-65.46 | 98.78 | 651.07 | < 0.001 |
| | 2 | 5 | 61.20 | 55.17-67.23 | 94.89 | 53.98 | < 0.001 |
| | 3 | 1 | 66.55 | 64.66-68.45 | NA | NA | NA |
| | 4 | 3 | 56.38 | 43.62-69.14 | 92.81 | 18.60 | < 0.001 |
| | 5 | 2 | 70.99 | 56.46-85.51 | 84.34 | 6.39 | 0.01 |
| Sample Size Frame | Large (> 1000) | 7 | 63.27 | 60.65-65.89 | 89.45 | 51.23 | < 0.001 |
| | Medium (400–1000) | 5 | 73.91 | 61.33-86.50 | 98.38 | 186.76 | < 0.001 |
| | Small (200–400) | 4 | 55.78 | 44.36-67.19 | 93.83 | 52.87 | < 0.001 |
| | Very Small (< 200) | 8 | 44.94 | 35.78-54.10 | 89.99 | 70.55 | < 0.001 |
| Study Year | 2010 and Before | 5 | 57.69 | 47.46-67.92 | 95.50 | 54.91 | < 0.001 |
| | 2011-2015 | 7 | 58.57 | 42.81-74.34 | 98.51 | 389.36 | < 0.001 |
| | 2016-2020 | 6 | 50.83 | 38.49-63.16 | 96.66 | 125.75 | < 0.001 |
| | After 2020 | 6 | 65.87 | 60.82-70.91 | 96.64 | 103.80 | < 0.001 |
| Healthcare Price Index | High | 9 | 57.38 | 45.92-68.85 | 98.06 | 392.22 | < 0.001 |
| | Medium | 7 | 55.64 | 42.34-68.94 | 99.20 | 281.82 | < 0.001 |
| | Low | 8 | 61.92 | 54.71-69.14 | 97.16 | 156.49 | < 0.001 |
| Primary Care Center Density | High | 5 | 61.96 | 44.96-78.96 | 99.15 | 241.40 | < 0.001 |
| | Medium | 3 | 52.40 | 31.69-73.10 | 99.17 | 71.38 | < 0.001 |
| | Low | 16 | 58.38 | 51.50-65.26 | 97.93 | 484.66 | < 0.001 |

**NA:** Not applicable.

* Refer to Supporting Information File 2 for information on each provinces region category.

## Discussion

This systematic review and meta-analysis aimed to provide an updated estimate of the prevalence of MetS among Iranian postmenopausal women. According to our results, MetS is prevalent in nearly three-fifths (≈ 58.5%) of postmenopausal women in Iran, suggesting a substantial burden of this condition in this demographic group. Notably, the pooled prevalence of MetS rose substantially to 64.10% in high-quality studies and 63.24% in population-based studies, suggesting that more rigorous methodologies yield higher estimates of the MetS burden. However, considerable statistical heterogeneity was observed, indicating significant variation between the included studies influenced by random error. Consequently, this pooled estimate should be viewed as an approximate summary of a broad distribution of prevalence rates rather than a precise national figure.

According to the NCEP ATP III guidelines, MetS is diagnosed when an individual has at least three of five specific components. These components are: elevated waist circumference (specific to sex and population), elevated triglycerides (≥150 mg/dL), reduced HDL cholesterol (<40 mg/dL in men, <50 mg/dL in women), elevated blood pressure (≥130/85 mmHg), and elevated fasting glucose (≥100 mg/dL). The presence of these factors signifies a cluster of cardiometabolic risks that significantly increase the likelihood of developing cardiovascular disease and type 2 diabetes.

Two prior meta-analyses by Ebtekar et al. [19] and Tabatabaei-Malazy et al. [20] estimated the prevalence of MetS among Iranian postmenopausal women at 51.6% and 58.8%, respectively, both lower than the rate identified in the current study. This discrepancy may be attributed to our inclusion of additional high-quality, population-based studies in the quantitative synthesis. Unlike Ebtekar et al. [19], our analysis revealed that both high-quality and population-based studies consistently reported higher MetS prevalence rates in this population.

Notably, the prevalence of MetS among Iranian postmenopausal women exceeds global estimates. Hallajzadeh et al. [14] reported a global MetS prevalence of approximately 36% in postmenopausal populations, substantially lower than Iranian rates. Similarly, studies from Poland [46], Korea [47,48], Japan [49], China [50,51], Canada [52], and the USA [53,54], indicate prevalence rates ranging from 20% to 40%. Even within the Middle East, Tabatabaei-Malazy et al. [20] documented a lower prevalence in Turkey (39%), Iran's northwestern neighbor. Despite their geographical proximity, this lower prevalence of MetS in Turkey compared to Iran, can be attributed to several key differences in lifestyle and dietary habits [55–57].

Turkish cuisine, particularly in coastal regions, incorporates more olive oil (rich in monounsaturated fats) compared to Iran's reliance on hydrogenated oils and animal fats, which are linked to dyslipidemia and insulin resistance [58,59]. Also, the Mediterranean diet includes more fish (omega-3 fatty acids) and fresh vegetables, which are protective against inflammation and MetS and have been shown to improve MetS and its components [60,61], whereas Iranian diets often feature more red meat and refined grains [62]. On the other hand, estimates from India and Saudi Arabia align more closely with Iranian data, reporting rates of 55% to 57% [63,64]. Similar dietary patterns (high refined carbs, unhealthy fats) and lifestyle factors (sedentary behavior, urbanization effects) may explain why Iran, India, and Saudi Arabia report higher MetS rates than Western or East Asian nations.

The significantly higher prevalence of MetS among Iranian postmenopausal women compared to global estimates and even some neighboring countries can be attributed to a combination of biological, lifestyle, socioeconomic, and healthcare-related factors. Traditional Iranian diets are often rich in refined and simple carbohydrates (e.g., white rice, bread), unhealthy fats (e.g., hydrogenated oils, ghee, fried foods), and low in fiber, which contributes to obesity, insulin resistance, and dyslipidemia, key components of MetS [59,65]. Also, Physical inactivity is rising among the Iranian population, particularly women, possibly due to cultural norms, lack of access to safe exercise spaces, and low awareness of the benefits of physical activity [66].

Rapid urbanization has led to more processed food consumption and reduced physical activity, increasing obesity and metabolic risks [20]. Pollution, vitamin D deficiency (due to conservative clothing norms), and other regional exposures may contribute to metabolic dysfunction [67]. Financial constraints may limit access to healthier food options (e.g., fresh

produce, lean proteins) and healthcare services, while limited health literacy and awareness of metabolic health risks may delay preventive measures [68,69]. Even when diagnosed, inconsistent use of lipid-lowering, antihypertensive, or antidiabetic drugs may worsen MetS progression.

Furthermore, our meta-regression analysis, consistent with prior evidence, demonstrated a positive association between advancing age and increased MetS prevalence among Iranian postmenopausal women. This age-dependent rate escalation reflects the synergistic interplay between menopausal hormonal changes and age-related metabolic deterioration, mediated through four possible pathways: 1. sarcopenia-induced reduction in basal metabolic rate, exacerbating obesity and insulin resistance [10]; 2. diminished oxidative capacity and mitochondrial dysfunction in aging tissues, impairing both lipid and glucose metabolism [70]; and 3. declining growth hormone and IGF-1 signaling, further disrupting metabolic homeostasis [71]. Also, while the potential cardiometabolic implications of menopausal timing are biologically plausible, with both early and late menopause potentially exacerbating MetS risk [72], our analysis found no significant associations. This null finding likely reflects insufficient statistical power, as only six studies provided adequate data on age at menopause.

Our subgroup and meta-regression analyses identified both study quality and study setting as considerable sources of heterogeneity, with high-quality and population-based studies yielding substantially higher pooled prevalence estimates (64.10% and 63.24%, respectively). This pattern can be explained by key methodological differences that affect both the selection of participants and the ascertainment of MetS.

The higher prevalence in high-quality studies likely reflects their more rigorous adherence to methodological standards. High-quality studies typically employed probability-based sampling methods (e.g., random sampling from well-defined population frames) rather than convenience sampling. This reduces selection bias and enhances the representativeness of the sample, ensuring that individuals with undiagnosed or subclinical MetS are included, thereby capturing the true community burden. Furthermore, high-quality studies may have demonstrated stricter and more consistent application of the NCEP ATP III diagnostic criteria, particularly in the standardized measurement of waist circumference and blood pressure, leading to more complete case ascertainment compared to studies with less rigorous protocols.

The difference based on study setting is equally telling. The higher prevalence found in population-based studies likely provides a more accurate reflection of the true community burden of MetS. These studies sample from a general population, capturing a wide spectrum of health states, including individuals who may have MetS but are not actively seeking care for its components. In contrast, institutional-based studies (e.g., those conducted in menopause clinics, diabetes clinics, or hospitals) are inherently susceptible to selection bias, as they sample from a pool of individuals who have already engaged with the healthcare system. This population is often "sicker" on average, but it can also be biased in the opposite direction; for instance, a study in a bone densitometry clinic might attract health-conscious individuals, potentially leading to an underestimation of the true population prevalence. The consistently lower and more variable prevalence in institutional settings (50.16%, 95%CI: 37.76–62.57) supports the notion that these estimates are highly context-dependent and less generalizable.

## Implications for clinical practice and policy

Given the high prevalence of MetS, routine screening for its components (abdominal obesity, hypertension, dyslipidemia, and hyperglycemia) should be integrated into standard care for postmenopausal women in Iran. Healthcare providers should prioritize regular blood pressure checks, anthropometric measurements, lipid profiles, and fasting glucose tests, within primary care facilities. Public health initiatives should focus on educating women about MetS risks after menopause, emphasizing preventive measures such as healthy eating and regular exercise. Collaboration between health ministries, educational institutions, and non-governmental organizations can amplify efforts to combat MetS through community-based programs. Nevertheless, policies regulating food industries to reduce trans fats and sugar in processed foods could help mitigate dietary risks.

Regarding the feasibility and implementation pathways within the Iranian context, a pragmatic approach is essential. A primary strategic opportunity lies in leveraging and strengthening the existing Iran Package of Essential Non-communicable (IraPEN) Diseases Interventions within the primary healthcare (PHC) network. MetS screening can be efficiently integrated into the routine care provided for middle-aged and older women attending health centers for other reasons (e.g., routine check-ups or management of other conditions). This opportunistic screening minimizes additional burden and leverages existing patient touchpoints. The screening protocol itself can be cost-effective, starting with simple anthropometric measurements (waist circumference, weight, height) and blood pressure checks, which require minimal resources. For women identified as at-risk, further investigation with fasting blood glucose and lipid profiles could be prioritized.

We acknowledge the resource constraints, including financial limitations and workforce capacity, particularly in underserved regions. To address this, implementation could be piloted in well-resourced provinces before nationwide scale-up. Furthermore, a task-sharing model, where trained community health workers ("*Behvarz*") conduct initial anthropometric and blood pressure measurements, could significantly reduce the workload on physicians and nurses. Digital health tools, such as integrated electronic health records within the PHC system, could facilitate patient tracking, remind healthcare providers of due screenings, and streamline data collection for monitoring and evaluation. Nevertheless, the high prevalence of MetS we found strongly justifies the cost and effort of implementing a screening program. The potential to prevent a significant number of future cases of cardiovascular disease and diabetes makes a compelling case for action.

## Limitations and future directions

While studies were conducted across multiple provinces, certain regions may be underrepresented (such as Sistan and Baluchestan, Zanjan, Lorestan, Khorasan, etc.), limiting generalizability to all Iranian postmenopausal women. We recommend that future studies prioritize underrepresented regions to ensure nationwide generalizability.

Many studies lacked detailed data on age at menopause, restricting meta-regression analysis. Researchers should systematically report key variables such as menopausal age, duration since menopause, and lifestyle factors (diet, physical activity) to facilitate more comprehensive meta-regression and subgroup analyses.

Furthermore, our analysis is subject to limitations related to the ascertainment of menopausal status. The definition of postmenopause across the included studies primarily relied on self-reported amenorrhea (≥12 months since the last menstrual period), which is standard in large epidemiological studies. However, this approach is susceptible to misclassification, as women with irregular cycles or those using hormonal therapies may inaccurately report their status.

Additionally, data on age at menopause were often unavailable or based on self-report, which is subject to recall bias. Such misclassification could have led to an underestimation or overestimation of the true MetS prevalence in a precisely defined postmenopausal population and may have contributed to the observed heterogeneity and the non-significant finding for age at menopause in our meta-regression. Future prospective studies with rigorous, standardized verification of menopausal status and age at onset would help mitigate this limitation.

Fourth, as previously noted, our analysis relied predominantly on the NCEP ATP III definition. A significant limitation of this criterion is its use of a universal waist circumference cutoff, which may lack sensitivity for the Iranian population. Given that Iranian and other Middle Eastern populations may exhibit a different adiposity patterns with higher metabolic risk at lower waist circumferences, the NCEP ATP III cutoff of ≥88 cm for women might lead to an underestimation of the true MetS burden [73,74]. Future national studies would benefit from developing and validating Iran-specific cutoffs for waist circumference to enhance the accuracy and clinical relevance of MetS diagnosis.

Additionally, despite efforts to reduce methodological heterogeneity, subgroup, and meta-regression analyses, significant unexplained statistical heterogeneity remains, suggesting other unmeasured factors influencing prevalence estimates. Future meta-analyses could employ advanced statistical methods or machine learning techniques to better account for confounding variables and improve the precision of pooled estimates.

Furthermore, while meta-regression identified participant age as a significant source of heterogeneity, a subgroup analysis by age was not feasible due to inconsistent and incomplete reporting of age data across primary studies. Future primary studies should report age-stratified prevalence data to enable more granular meta-analyses. Also, as is common in meta-epidemiology, the number of available studies for each covariate analysis was limited, which reduces the statistical power and increases the risk of both Type I and Type II errors. Consequently, the significant associations should be considered exploratory and indicative of potential influences rather than definitive proof of causality. Similarly, the non-significant findings for other covariates, such as age at menopause, may reflect a genuine lack of association or, alternatively, an insufficient number of studies to detect one. Therefore, these results are best viewed as generating hypotheses to be tested in future primary studies with larger, more diverse populations and in updated meta-analyses as the evidence base grows.

Also, we detected evidence of potential publication bias. It is plausible that smaller studies reporting non-significant or lower prevalence rates of MetS remain unpublished. Nevertheless, the robustness of our main estimate was confirmed by a sensitivity analysis, and the trim-and-fill method indicated that adjusting for potential missing studies would not substantially change the pooled prevalence. This reinforces the conclusion that MetS is highly prevalent in this demographic, even when accounting for this potential bias

Finally, to better understand the evolving burden of MetS and disentangle the complex interplay of lifestyle, environmental, and policy changes, future research should prioritize longitudinal studies that track incidence and progression, and conduct time-stratified analyses of population-level data to examine specific temporal trends.

## Conclusion

This study highlights a high prevalence of MetS (almost three-fifths of Iranian postmenopausal women), a burden that rises to over 63% in high-quality, population-based studies. The high prevalence underscores the urgent need to integrate routine MetS screening into the standard care protocol for middle-aged and postmenopausal women within Iran's primary healthcare network starting with simple anthropometric and blood pressure measurements. To ensure feasibility, a task-sharing model utilizing trained community health workers (*Behvarz*) for initial screenings can expand reach and reduce the burden on physicians, especially in underserved areas. These findings compellingly argue for the establishment of longitudinal monitoring systems to track trends and the urgent implementation of coordinated, national-level interventions.

## Supporting information

**S1 File. Search query in PubMed, Scopus, Embase, and Web of Science.**
(DOCX)

**S2 File. Definition of regions and categories.**
(DOCX)

**S3 File. Quality assessment details using JBI checklist.**
(DOCX)

**S4 File. Figures related to subgroup analyses of included papers.**
(DOCX)

## Acknowledgments

The authors would like to thank the Clinical Research Development Unit of Imam Khomeini Hospital, Mazandaran University Of Medical Science Sari, Iran for their support, cooperation, and assistance throughout the period of study.

## Author contributions

**Conceptualization:** Erfan Ghadirzadeh, Mojgan Geran, Morteza Biabani, Mobina Gheibi, Maryam Zarrinkamar.

**Data curation:** Erfan Ghadirzadeh, Atoosa Mahmoodi, Pedram Nezhadnaderi, Anita Ziari, Morteza Biabani.

**Formal analysis:** Mahmood Moosazadeh.

**Methodology:** Erfan Ghadirzadeh.

**Project administration:** Erfan Ghadirzadeh, Mojgan Geran, Mobina Gheibi, Maryam Zarrinkamar.

**Resources:** Atoosa Mahmoodi, Naseh Yousefi, Pedram Nezhadnaderi.

**Software:** Erfan Ghadirzadeh, Mahmood Moosazadeh.

**Supervision:** Mojgan Geran, Mobina Gheibi, Mahmood Moosazadeh, Maryam Zarrinkamar.

**Validation:** Erfan Ghadirzadeh, Naseh Yousefi, Mojgan Geran, Mobina Gheibi.

**Visualization:** Erfan Ghadirzadeh, Atoosa Mahmoodi, Mahmood Moosazadeh.

**Writing – original draft:** Erfan Ghadirzadeh.

**Writing – review & editing:** Erfan Ghadirzadeh, Atoosa Mahmoodi, Naseh Yousefi, Pedram Nezhadnaderi, Mojgan Geran, Anita Ziari, Morteza Biabani, Mobina Gheibi, Mahmood Moosazadeh, Maryam Zarrinkamar.

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
