## [Decision Letter · Decision Letter 0]

1 Sep 2025

Dear Dr. Zarrinkamar,

Thank you for submitting your manuscript to PLOS ONE. After careful consideration, we feel that it has merit but does not fully meet PLOS ONE’s publication criteria as it currently stands. Therefore, we invite you to submit a revised version of the manuscript that addresses the points raised during the review process.

**ACADEMIC EDITOR: **

We look forward to receiving your revised manuscript.

Kind regards,

Ozra Tabatabaei-Malazy

Academic Editor

PLOS ONE

Journal Requirements:

Reviewers' comments:

Reviewer's Responses to Questions

**Comments to the Author**

1. Is the manuscript technically sound, and do the data support the conclusions?

Reviewer #1: Yes

Reviewer #2: Yes

Reviewer #3: Partly

Reviewer #4: Yes

2. Has the statistical analysis been performed appropriately and rigorously?

Reviewer #1: Yes

Reviewer #2: No

Reviewer #3: Yes

Reviewer #4: Yes

3. Have the authors made all data underlying the findings in their manuscript fully available?

Reviewer #1: Yes

Reviewer #2: Yes

Reviewer #3: Yes

Reviewer #4: Yes

4. Is the manuscript presented in an intelligible fashion and written in standard English?

Reviewer #1: No

Reviewer #2: Yes

Reviewer #3: No

Reviewer #4: Yes

Reviewer #1: This remarkable and comprehensive work reports on the far-reaching consequences of metabolic syndrome in women after they enter the monophase, because it represents the main burden in the field of NCDs on the different dimensions of health. The results can highlight the importance of this risk factor and its potential impact on women after the fertility period in Iran. I have made a few comments on this manuscript, consideration of which may improve the quality and clarity of your reports:

In the abstract, you should specify the exact time frame for the literature and database search, e.g. (August 2010 to April 2025).

In the introduction, in lines 54 and 55, you mentioned the Burdens of MetS in Iran, for more clarity you need to give the year of this rate and statistics.

Two more sentences about the components and criteria for MetS in the first paragraph of the discussion could be informative for readers and researchers.

The exclusion criteria (number 4) for study selection should be rewritten and simplified, e.g. what you mean by “overlapping population”

In the data extraction section (line;146), “number of postmenopausal females” should be replaced by “proportion of females….”.

In the section on participants, you should specify the minimum age of female participants or the age of participants, as the start of the monophase in different population groups.

Regarding the significant contribution of participant age to heterogeneity, a subgroup analysis based on age has been suggested to improve the significance of the pooled prevalence you report.

The periods of the studies should also be considered in your subgroup analysis. Because if there is a trend in prevalence over time, it should be discussed further.

Reviewer #2: Dear authors,

The manuscript highlight the need for high quality evidence regarding the prevalence of metabolic syndrome in postmenopausal women in Iran - a subset of the general population which is often neglected despite an optimal availability for delivery primary primary health care services. The following recommendations for revisions are made in hopes that the final manuscript may better address the study aims:

1. A significant amount of time has lapsed since the previously mentioned systematic reviews of prevalence, and results from the PERSIAN cohort - although of high epidemiological value - are not the sole reason for the need of an updated appraisal of the evidence. Please highlight other newly published evidence in the introduction.

2. Please provide further justifications for targeting the prevalence of MetS in postmenopausal women in the manuscript introduction. Particularly, evidence regarding coverage of related services, control, and under-diagnosis - if applicable. Furthermore the introduction would benefit from inclusion of specific policies the current update might impact would impact.

3. Retrospective assurance of the inclusion of all studies included in previous review on this topic and the evaluation of each study's citing and cited literature is recommended for a comprehensive review - if not already performed.

4. Recent evaluations have shown possible fundamental errors in interpretation caused by the use of the LFK index as a measure of small-study bias. The use of Egger's test with a higher than conventional threshold of significance (< 0.1 or 0.2 depending on the number of included studies) might be more appropriate. Alternatively selection models may be employed to evaluate for publication bias. The heterogeneity found in prevalence reports from different studies may additionally be a significant factor in identification of small-study bias via plot asymmetry approaches. Smaller studies may indeed be affected by sampling bias (bearing in mind that these studies are usually not population-based) leading to skewed estimates. The suggestion of publication bias when the target effect is prevalence might in turn be less expected as higher prevalence reports might not directly motivate publication. Further evaluation of such a possibility and its incorporation into the manuscript discussion would be appreciated.

- Schwarzer G, Rücker G, Semaca C. LFK index does not reliably detect small-study effects in meta-analysis: A simulation study. Res Synth Methods. 2024 Jul;15(4):603-615. doi: 10.1002/jrsm.1714. Epub 2024 Mar 11. PMID: 38467140.

5. Interpretation of meta-regression results when the number if included studies is insufficient (<10 per adjusted variable) might be subject to type II errors. Interpretations arising from subgroup analyses may be more appropriate under such circumstances. For example the proposition that age has no significant effect on the prevalence of MetS seems to be both counterintuitive and not supported by the evidence. Additionally, all subgroups evaluated have also demonstrated a high degree of heterogeneity (inconsistency) in their respective estimates. Therefore other possible sources of heterogeneity should be explored or discussed in review limitations. One such variables to explore are changes in MetS prevalence according to the timeframe of the included studies - as the identified heterogeneity may be in fact indicative of underlying trends in the prevalence of MetS. More recent reports may have similarly benefited from better methodology and reporting leading to the proposition of "higher prevalence in high-quality studies". At minimum a subgroup analysis stratifying studies by study time-frame seems to be warranted.

6. The review title highlight the PRISMA compliance of the manuscript. Appraisal of the quality of evidence (certainty) is an oft-neglected and essential step in ensuring PRISMA compliance. Although the broadly used GRADE approach may not be optimized for meta-analyses of prevalence, several modifications/alternatives may be explored. Use of GRADE with ad hoc modifications is similarly not without merit.

- Yousefifard M, Shafiee A. Should the reporting certainty of evidence for meta-analysis of observational studies using GRADE be revisited? Int J Surg. 2023 Feb 1;109(2):129-130. doi: 10.1097/JS9.0000000000000114.

- Borges Migliavaca C, Stein C, Colpani V, Barker TH, Munn Z, Falavigna M; Prevalence Estimates Reviews – Systematic Review Methodology Group (PERSyst). How are systematic reviews of prevalence conducted? A methodological study. BMC Med Res Methodol. 2020 Apr 26;20(1):96. doi: 10.1186/s12874-020-00975-3.

Kind regards,

Reviewer

Reviewer #3: This manuscript explores an important topic in women’s health by systematically reviewing and quantifying the prevalence of metabolic syndrome (MetS) among Iranian postmenopausal women. The inclusion of recent data from large-scale cohorts and the effort to update and improve on prior reviews are notable strengths. Additionally, the use of subgroup and meta-regression analyses to explore heterogeneity adds analytical depth. However, there are several areas where revisions are necessary to improve clarity, interpretability, and overall scientific rigor.

Abstract

• Improve clarity by explicitly stating the role of study quality, setting, and age as sources of heterogeneity.

• The conclusion could extend beyond screening to mention broader public health or preventive implications.

Introduction

• The background is comprehensive but could be more concise and focused on the rationale for this study.

• Better articulation is needed regarding how this study improves upon previous reviews. Avoid simply citing inclusion of the PERSIAN cohort—highlight methodological differences more explicitly.

Methods

• The POLIS criteria should be defined more clearly and not just listed as an acronym.

• While the decision to prioritize the NCEP ATP III definition is acceptable, a brief note on its limitations (e.g., generalizability or sensitivity) would strengthen the justification.

• Clarify whether study selection and data extraction were done independently by two reviewers.

• The scoring thresholds used for the JBI tool (e.g., 7–9 = high quality) should be referenced or justified.

Results

• The pooled prevalence estimate is clear, but the interpretation should be more cautious given the extreme heterogeneity (I² > 98%).

• While sensitivity and publication bias analyses were conducted appropriately, their implications deserve more emphasis.

• Figure 2 (forest plot) needs to be regenerated in higher resolution. Font size and axis clarity are insufficient for publication.

• In Table 2, regional labels (Region 1–5) should be linked clearly to Supplementary File 2 or relabeled more meaningfully.

Discussion

• The discussion covers relevant comparisons across countries and possible mechanisms, but some sections are overly speculative (e.g., attributing prevalence differences to national dietary patterns). These need to be rewritten with more caution and supporting evidence.

• The biological discussion, while accurate, is too detailed in parts and detracts from the epidemiological focus of the paper.

• The interpretation of meta-regression findings should better acknowledge the limitations of covariate-level analyses with small numbers of studies.

• Greater attention should be given to why study quality and setting were associated with different prevalence estimates.

Implications and Limitations

• The suggestion to incorporate routine screening into clinical practice is reasonable, but discussion of real-world feasibility, resource constraints, and specific implementation pathways in the Iranian context would add value.

• The limitations section is appropriate but could be expanded to discuss misclassification of menopausal status or potential biases in self-reported age at menopause.

• The time span across studies may reflect secular trends in MetS prevalence, which is not fully addressed.

Language and Presentation

• Numerous grammatical and stylistic issues are present throughout. The manuscript would benefit significantly from professional English editing.

• Figures and tables require formatting improvements for consistency and legibility.

Conclusion

• The conclusion appropriately summarizes the main findings but could be expanded slightly to mention the need for targeted public health policies or future longitudinal research.

Reviewer #4: 1. In the introduction, it would be helpful to also provide a global perspective on the issue. This way, we can see it is particularly important in Iran compared with the global context.

2. In the Methods section, it is mentioned that one of the eligible study designs was clinical trial. Could you please explain this.

3. In the table, there are some abbreviations that do not seem to be commonly used. Please write the full term instead. (for example, instead of 'AAM,' write 'mean menopausal age). Also, please mention the units for all variables and use a consistent number of decimal places throughout the table.

4. Please clarify the institution based and public based definitions in methods section. Also briefly discuss the five regions of Iran in methods section.

**Do you want your identity to be public for this peer review?** For information about this choice, including consent withdrawal, please see our Privacy Policy

Reviewer #1: No

Reviewer #2: No

Reviewer #3: No

Reviewer #4: No

---

## [Author Response · Author response to Decision Letter 1]

3 Oct 2025

Dear PLoS One Editorial Team,

We would like to express our sincere gratitude to you and your team of reviewers for the time and consideration you gave to our manuscript titled “Prevalence of Metabolic Syndrome among Iranian Postmenopausal Females: a Systematic Review and Meta-Analysis”. Your expertise and attention to detail were invaluable in helping us to refine our research and present it in a way that is clear and meaningful. We are especially grateful for the helpful comments and feedbacks provided by the reviewer. Reviewer’s insightful critiques and suggestions allowed us to improve the quality of our research and strengthen the overall presentation of our work. We appreciate the opportunity you have given us to share our research with the academic community. Below are the answers to your comments. Furthermore, changes in the manuscript have been highlighted in yellow. We hope you find it helpful.

Editor

Comment: Please revise Flowchart in accordance with PRISMA 2020 guideline.

Answer: Thank you for your feedback. We have revised this flowchart.

Reviewer #1

Comment: This remarkable and comprehensive work reports on the far-reaching consequences of metabolic syndrome in women after they enter the monophase, because it represents the main burden in the field of NCDs on the different dimensions of health. The results can highlight the importance of this risk factor and its potential impact on women after the fertility period in Iran. I have made a few comments on this manuscript, consideration of which may improve the quality and clarity of your reports:

In the abstract, you should specify the exact time frame for the literature and database search, e.g. (August 2010 to April 2025).

Answer: Thank you for your comment. Databases were searched from their dates of inception until April 2025. This has been specified in the revised abstract.

Comment: In the introduction, in lines 54 and 55, you mentioned the Burdens of MetS in Iran, for more clarity you need to give the year of this rate and statistics.

Answer: Thank you for your feedback. This statistic was the result of an updated meta-analysis by Moradkhani et al. which was published on 2025. This has been indicated on line 58.

Comment: Two more sentences about the components and criteria for MetS in the first paragraph of the discussion could be informative for readers and researchers.

Answer: The authors gratefully acknowledge the reviewer's suggestion. Although it is standard practice for the opening paragraph of the Discussion to summarize the principal findings, we agree that providing 2-3 sentences on the components and diagnostic criteria for MetS would enhance the manuscript's clarity for readers. A paragraph to this effect has therefore been added as the second paragraph of the Discussion section.

Comment: The exclusion criteria (number 4) for study selection should be rewritten and simplified, e.g. what you mean by “overlapping population”

Answer: We thank the reviewer for this valuable suggestion. To improve clarity and precision, we have rewritten the fourth exclusion criterion in the manuscript. This criterion explicitly prevents the inclusion of multiple reports from the same parent study or cohort, thereby avoiding the double-counting of participants and the associated statistical bias of multiplicity in the meta-analysis.

Comment: In the data extraction section (line;146), “number of postmenopausal females” should be replaced by “proportion of females….”.

Answer: Thank you for your feedback. This has been rewritten.

Comment: In the section on participants, you should specify the minimum age of female participants or the age of participants, as the start of the monophase in different population groups.

Answer: We thank the reviewer for this insightful comment. The reviewer rightly points out that the natural age at menopause onset can vary across populations. In our study, the operational definition for a "postmenopausal female" was explicitly based on the clinical criterion of "cessation of menstrual period for at least 12 months" (as stated in the Participants section), rather than on a specific age threshold. This definition accurately identifies the postmenopausal state regardless of the population's average age at menopause. However, to fully address the reviewer's concern and provide greater methodological clarity, we have now added the following sentence to the Participants subsection in the revised manuscript:

"This definition based on amenorrhea, rather than a predetermined age cutoff, was employed to accurately capture the postmenopausal status across all included studies, accounting for potential variations in the natural age at menopause within the Iranian population."

Furthermore, we have extracted and presented the mean chronological age and menopausal age of participants from each included study (if reported) in Table 1, which provides readers with a picture of the age profile of the synthesized population.

Comment: Regarding the significant contribution of participant age to heterogeneity, a subgroup analysis based on age has been suggested to improve the significance of the pooled prevalence you report.

Answer: We thank the reviewer for this valuable suggestion. We agree that a subgroup analysis based on age would be highly informative. As the reviewer noted, our meta-regression confirmed that participant age was a significant source of heterogeneity (P: 0.038). However, performing a reliable subgroup analysis was precluded by a critical data limitation in the primary studies. Specifically, only 16 of the 24 included studies reported the mean age of their postmenopausal participants. Among these, the mean age ranged from 53.98 to 69.14 years. Categorizing these studies into discrete age groups (e.g., <55 vs. ≥55 years) would be methodologically unsound, as the minimum and maximum ages and the underlying age distributions across these studies are unknown and likely overlap significantly.

To address this important limitation transparently, we have now added the following statement to the "Limitations and Future Directions" section of the revised manuscript:

"Furthermore, while meta-regression identified participant age as a significant source of heterogeneity, a subgroup analysis by age was not feasible due to inconsistent and incomplete reporting of age data across primary studies. Future primary studies should report age-stratified prevalence data to enable more granular meta-analyses."

We acknowledge that an Individual Participant Data (IPD) meta-analysis would be the ideal approach to overcome this limitation, though it is often not feasible. We believe that explicitly stating this limitation in the manuscript, as now done, appropriately highlights the issue and guides future research, as the reviewer intended.

Comment: The periods of the studies should also be considered in your subgroup analysis. Because if there is a trend in prevalence over time, it should be discussed further.

Answer: Thank you for your suggestion. We have added subgroup analysis results based on study year (categorized as studies on and before 2010, 2011 to 2015, 2016 to 2020, and after 2020) which did not yield specific trends in prevalence over time.

Reviewer #2

Dear authors,

The manuscript highlight the need for high quality evidence regarding the prevalence of metabolic syndrome in postmenopausal women in Iran - a subset of the general population which is often neglected despite an optimal availability for delivery primary primary health care services. The following recommendations for revisions are made in hopes that the final manuscript may better address the study aims:

Comment 1. A significant amount of time has lapsed since the previously mentioned systematic reviews of prevalence, and results from the PERSIAN cohort - although of high epidemiological value - are not the sole reason for the need of an updated appraisal of the evidence. Please highlight other newly published evidence in the introduction.

Answer: We thank the reviewer for this insightful comment. We completely agree that the need for an updated synthesis extends beyond the inclusion of the PERSIAN cohort. To address this point thoroughly, we have revised the Introduction to more comprehensively outline the gaps left by previous reviews and the breadth of new evidence now available.

Since the last major reviews in 2018, a substantial number of new studies have been published. We now mention that our search identified several recent, high-quality population-based cohorts (such as the Tabari, Hoveyzeh, Guilan, and Fasa Cohort Studies, among others) which were not available for prior analyses. Also, our review employs more rigorous and contemporary methodological standards (e.g., comprehensive search, subgroup analysis, meta-regression) to address the heterogeneity that limited earlier efforts. Below, we present to you several advantages of our paper compared to prior reviews in this field, and outline the most crucial limitations of each previous review and demonstrate how our study enhances methodological rigor and contributes to advancing the field. These were previously mentioned in the Cover Letter during the submission:

o Review No 1. Ebtekar F, Dalvand S, Gheshlagh RG. The prevalence of metabolic syndrome in postmenopausal women: A systematic review and meta-analysis in Iran. Diabetes & Metabolic Syndrome: Clinical Research & Reviews. 2018 Nov 1;12(6):955-60.

o Issue 1: Suboptimal Search Strategy and Syntax

Search strategy and syntax are not optimal. We found several key terms which were missing in the search syntax such as cardiometabolic syndrome, etc.

o Issue 2: Inconsistency in Diagnostic Criteria and Subgroup Analysis

Although the previous study acknowledges variability in diagnostic criteria across primary studies and attempts to address this through subgroup analysis, such an approach inadvertently introduces methodological heterogeneity. Upon re-evaluating the included studies, we observed that many of them reported data based on the NCEP ATP III criteria, even when multiple definitions (e.g., IDF, JIS) were cited. By consistently extracting metabolic syndrome status using the NCEP ATP III criteria, our analysis minimizes methodological heterogeneity, thereby enhancing the reliability and comparability of the findings.

o Issue 3: Missing Key Relevant Studies

A critical examination of the previous meta-analysis reveals the exclusion of several seminal studies in this field, including Delavar (2009), Shahvazi (2016), Sadat (2015), and etc. These omissions represent a significant limitation, as these studies contain pertinent epidemiological data on metabolic syndrome prevalence in the target population. The absence of these key publications may have substantially influenced the pooled estimates and overall conclusions of the previous analysis.

o Issue 4: Failure to Address Population Overlap in Included Studies

A significant methodological concern in the previous meta-analysis is the inclusion of studies with overlapping populations without appropriate adjustment. Notably, the analysis incorporated multiple publications from the Tehran Lipid and Glucose Study at enrollment phase (e.g., Ziaei 2011 and Ziaei 2013; Farahmand 2013, Farahmand 2014, and Ayni 2007; Jouyandeh 2013 and Namazi Shabestari 2016) that essentially represent duplicate population samples. This oversight violates the fundamental assumption of independence required for valid meta-analytic pooling and may have artificially inflated the sample size while introducing bias in the effect estimates.

o Issue 5: Failure to Account for Multiplicity in Data Synthesis

The previous meta-analysis committed a critical methodological error by incorporating duplicate data from the same study population multiple times in their synthesis. A clear example of this multiplicity occurs with the Maharlouei et al. (2013) study, where prevalence estimates derived from different diagnostic criteria (NCEP ATP III and IDF) for the identical population were treated as independent datasets. This approach artificially inflates the sample size and violates the fundamental statistical principle of data independence, potentially distorting the pooled estimates and their precision.

o Issue 6: Lack of Transparency in Quality Assessment Methodology

The previous meta-analysis suffers from insufficient methodological transparency in its quality appraisal process. While the authors classified studies as high, medium, or low quality, they failed to provide supplementary materials documenting the rationale for individual study ratings. This opaque approach prevents critical evaluation of the quality assessment's validity and reproducibility.

o Issue 7: Inconsistent Menopause Definition

The prior meta-analysis failed to address significant heterogeneity in the operational definition of postmenopausal status across included studies. While some investigations defined menopause as ≥12 months of amenorrhea (consistent with WHO criteria), others employed more stringent thresholds (e.g., ≥3 years). Notably, our analysis revealed that studies reporting extended definitions also provided disaggregated data for the standard 12-month threshold.

Overall advantages and strengths of our manuscript compared to Ebtekar et al:

In contrast to the study by Ebtekar et al., our research employed a more robust search strategy, incorporating key terms derived from both MeSH and free-text methods. Furthermore, we expanded our search to include over 10 local and international databases and search engines, enabling the identification of several studies overlooked in their work. We also observed that nearly all included studies assessed metabolic syndrome status using either the NCEP ATP III criteria exclusively or in combination with other diagnostic criteria. To minimize methodological heterogeneity, a factor not addressed by Ebtekar et al., we exclusively extracted data based on the NCEP ATP III definition. Additionally, we meticulously evaluated eligible studies to prevent multiplicity and population overlap. To ensure methodological rigor, we performed a quality assessment using the internationally recognized Joanna Briggs Institute critical appraisal checklist for prevalence studies and provided full results in a supplementary file to enhance transparency. Moreover, we addressed methodological heterogeneity in the definition of menopause by consolidating varying criteria, resulting in a broadly defined postmenopausal population characterized by the cessation of menstruation for at least 12 consecutive months.

o Review No 2: Tabatabaei-Malazy O, Djalalinia S, Asayesh H, Shakori Y, Esmaeili Abdar M, Mansourian M, Mahdavi Gorabi A, Noroozi M, Qorbani M. Menopause and metabolic syndrome in the Middle East countries; a systematic review and meta-analysis study. Journal of diabetes & metabolic disorders. 2018 Dec 31;17:357-64.

o Issue 1: Suboptimal Search Strategy and Syntax

Search strategy and syntax are not optimal. Also, search was conducted on only three databases. We found several key terms which were missing in the search syntax such as cardiometabolic syndrome, etc.

o Issue 2: Absence of Sensitivity Analysis

o Issue 3: Lack of Publication Bias Assessment

o Issue 4: Insufficient Exploration of Heterogeneity

Overall advantages and strengths of our manuscript compared to Tabatabaei-Malazy et al:

In contrast to the study by Tabatabaei-Malazy et al., our research employed a more robust search strategy, incorporating key terms derived from both MeSH and free-text methods. Furthermore, we expanded our search to include over 10 local and international databases and search engines, enabling the identification of several studies overlooked in their work. Moreover, we: 1. conducted sensitivity analysis by leave-one-out method to assess individual study influence, 2. implemented a comprehensive publication bias assessment protocol including visual inspection of Doi plots (and the LFK index), Egger and Begg’s tests, and trim & fill method, 3. advanced the results through subgroup analysis and meta-regression.

We believe these revisions and answers to your comment provide a much stronger and more complete justification for our study, acc

---

## [Decision Letter · Decision Letter 1]

5 Nov 2025

Dear Dr. Zarrinkamar,

Thank you for submitting your manuscript to PLOS ONE. After careful consideration, we feel that it has merit but does not fully meet PLOS ONE’s publication criteria as it currently stands. Therefore, we invite you to submit a revised version of the manuscript that addresses the points raised during the review process.

We look forward to receiving your revised manuscript.

Kind regards,

Ozra Tabatabaei-Malazy

Academic Editor

PLOS ONE

Journal Requirements:

Reviewers' comments:

Reviewer's Responses to Questions

**Comments to the Author**

Reviewer #1: All comments have been addressed

Reviewer #2: All comments have been addressed

Reviewer #3: (No Response)

Reviewer #4: All comments have been addressed

2. Is the manuscript technically sound, and do the data support the conclusions?

Reviewer #1: Yes

Reviewer #2: Yes

Reviewer #3: Yes

Reviewer #4: Yes

3. Has the statistical analysis been performed appropriately and rigorously?

Reviewer #1: Yes

Reviewer #2: Yes

Reviewer #3: Yes

Reviewer #4: Yes

4. Have the authors made all data underlying the findings in their manuscript fully available?

Reviewer #1: Yes

Reviewer #2: Yes

Reviewer #3: Yes

Reviewer #4: Yes

5. Is the manuscript presented in an intelligible fashion and written in standard English?

Reviewer #1: Yes

Reviewer #2: Yes

Reviewer #3: Yes

Reviewer #4: Yes

Reviewer #1: Thank you for sharing a revised and improved manuscript. It now has scientific and methodological quality for the insight of other researchers.

Reviewer #2: The manuscript revised perfectly, and all revisions are made. The revised version should be accepted ..................................................................................................................................................

Reviewer #3: The authors have made clear improvements in response to earlier feedback. The revised manuscript is more coherent, methodologically transparent, and analytically detailed, especially regarding heterogeneity and the influence of study quality and setting. Nevertheless, several important issues remain that should be addressed to strengthen interpretability, contextual depth, and overall presentation quality before publication.

• How might the authors expand the conclusion to better describe how these findings inform public health or preventive strategies in Iran?

• What are the main limitations of the NCEP ATP III definition in this context—particularly regarding sensitivity and applicability to Middle Eastern populations?

• Given the very high heterogeneity (I² = 98.6%), should the pooled prevalence be interpreted more cautiously as an approximate national estimate rather than a precise figure?

• Are there plans to improve the resolution and legibility of the figures, especially the forest and Doi plots, for publication quality?

• Can the biological explanation in the Discussion be further streamlined to maintain focus on epidemiological interpretation and variability across studies?

• What factors might explain the association between study quality, study setting, and prevalence estimates—such as sampling design, representativeness, or analytical rigor?

• Would it be possible to include a brief discussion on the feasibility of implementing routine MetS screening in Iran, considering resource constraints or access challenges?

• Have the authors considered potential misclassification of menopausal status and bias in self-reported age at menopause, and how these might influence subgroup or meta-regression findings?

• Could the temporal span of the included studies have introduced secular trends in lifestyle or healthcare access that partly explain heterogeneity?

• Might it be helpful to mention that future research could use longitudinal or time-stratified analyses to explore temporal changes in MetS prevalence?

• Will the authors review tables and figures for consistent formatting and ensure sufficient clarity in labels and legends?

• Is another round of professional language editing planned to address the remaining minor grammatical inconsistencies?

• Finally, would the conclusion benefit from a short statement emphasizing the need for longitudinal monitoring or national-level interventions to address the high burden of MetS among postmenopausal women?

Reviewer #4: Thank you to the authors for addressing the previous comments and revising the manuscript accordingly. The revisions have improved the clarity and quality of the paper. I have no further concerns.

**Do you want your identity to be public for this peer review?** For information about this choice, including consent withdrawal, please see our Privacy Policy

Reviewer #1: No

Reviewer #2: No

Reviewer #3: No

Reviewer #4: No

---

## [Author Response · Author response to Decision Letter 2]

6 Nov 2025

Dear PLoS One Editorial Team,

We would like to express our sincere gratitude to you and your team of reviewers for the time and consideration you gave to our manuscript titled “Prevalence of Metabolic Syndrome among Iranian Postmenopausal Females: a Systematic Review and Meta-Analysis”. Your expertise and attention to detail were invaluable in helping us to refine our research and present it in a way that is clear and meaningful. We are especially grateful for the helpful comments and feedbacks provided by the reviewer. Reviewer’s insightful critiques and suggestions allowed us to improve the quality of our research and strengthen the overall presentation of our work. We appreciate the opportunity you have given us to share our research with the academic community. Below are the answers to your comments. Furthermore, changes in the manuscript have been highlighted in yellow. We hope you find it helpful.

Reviewer #3:

The authors have made clear improvements in response to earlier feedback. The revised manuscript is more coherent, methodologically transparent, and analytically detailed, especially regarding heterogeneity and the influence of study quality and setting. Nevertheless, several important issues remain that should be addressed to strengthen interpretability, contextual depth, and overall presentation quality before publication.

• How might the authors expand the conclusion to better describe how these findings inform public health or preventive strategies in Iran?

Reply: Thank you for your comment. We have revised the conclusion section with specific instructions on how our findings can be translated into public health practice. The reported prevalence rates in our results informs public health policy makers of the MetS burden in this population in order to consider any possible changes in current preventive strategies (which is another title to be studied).

• What are the main limitations of the NCEP ATP III definition in this context—particularly regarding sensitivity and applicability to Middle Eastern populations?

Reply: Thank you for your comment. We have added a paragraph to the Limitation section of the manuscript discussing this matter (line 451-458).

• Given the very high heterogeneity (I² = 98.6%), should the pooled prevalence be interpreted more cautiously as an approximate national estimate rather than a precise figure?

Reply: Thank you for your feedback. The considerable statistical heterogeneity (I² = 98.6%) indicates that the included studies show substantial variation in their individual prevalence estimates, which are influenced by random error. Therefore, the pooled prevalence of 58.42% should indeed not be interpreted as a single, precise national figure but rather as a robust and informative summary estimate of a wide and variable range. This has been mentioned on line 312-316.

• Are there plans to improve the resolution and legibility of the figures, especially the forest and Doi plots, for publication quality?

Reply: Thank you for your feedback. We have exported the images with highest possible quality and resolution from the STATA software.

• Can the biological explanation in the Discussion be further streamlined to maintain focus on epidemiological interpretation and variability across studies?

Reply: Thank you for your suggestion. These biological explanations were relocated to Introduction. The goal was to condense the detailed pathophysiology and directly link it to the "why" behind our study aims. We have further streamlined the biological explanations accordingly.

• What factors might explain the association between study quality, study setting, and prevalence estimates—such as sampling design, representativeness, or analytical rigor?

Reply: Thank you for your comment. We have already discussed this on line 379-399.

• Would it be possible to include a brief discussion on the feasibility of implementing routine MetS screening in Iran, considering resource constraints or access challenges?

Reply: Thank you for your suggestion. We have already discussed this on line 411-430.

• Have the authors considered potential misclassification of menopausal status and bias in self-reported age at menopause, and how these might influence subgroup or meta-regression findings?

Reply: Thank you for your comment. This issue has already been mentioned on line 440-450.

• Could the temporal span of the included studies have introduced secular trends in lifestyle or healthcare access that partly explain heterogeneity?

Reply: Thank you for your suggestion. Our meta-regression found no significant association between study year and prevalence estimates (P = 0.521). Additionally, a sub-group analysis was performed on study year which indicated no reduction in statistical heterogeneity. These null findings suggest that the effect of temporal trends is likely non-linear and is powerfully confounded by other variables. The strong, independent associations we found for study quality and setting indicate that methodological differences between studies are a more dominant driver of heterogeneity than any underlying temporal trend, effectively masking its signal. Therefore, while secular trends in lifestyle and healthcare are undoubtedly important at a population level, their net effect on our pooled estimate is inconsistent and obscured by the substantial methodological variation across the Iranian research landscape.

• Might it be helpful to mention that future research could use longitudinal or time-stratified analyses to explore temporal changes in MetS prevalence?

Reply: Thank you for your comment. We discussed this on line 482-485.

• Will the authors review tables and figures for consistent formatting and ensure sufficient clarity in labels and legends?

Reply: Thank you for your feedback. We have rechecked tables and figures for typographical errors and made necessary amendments. We have also exported the highest exportable resolution for images from STATA.

• Is another round of professional language editing planned to address the remaining minor grammatical inconsistencies?

Reply: Thank you for your comment. We have proofread the manuscript with an English language editor and made amendments where necessary.

• Finally, would the conclusion benefit from a short statement emphasizing the need for longitudinal monitoring or national-level interventions to address the high burden of MetS among postmenopausal women?

Reply: Thank you for your comment. We have integrated this into the conclusion section.

---

## [Decision Letter · Decision Letter 2]

25 Nov 2025

Prevalence of Metabolic Syndrome among Iranian Postmenopausal Females: a Systematic Review and Meta-Analysis

PONE-D-25-37457R2

Dear Dr. Zarrinkamar,

We’re pleased to inform you that your manuscript has been judged scientifically suitable for publication and will be formally accepted for publication once it meets all outstanding technical requirements.

Kind regards,

Ozra Tabatabaei-Malazy

Academic Editor

PLOS ONE

Additional Editor Comments (optional):

Reviewers' comments:

Reviewer's Responses to Questions

**Comments to the Author**

Reviewer #3: All comments have been addressed

2. Is the manuscript technically sound, and do the data support the conclusions?

Reviewer #3: Yes

3. Has the statistical analysis been performed appropriately and rigorously?

Reviewer #3: Yes

4. Have the authors made all data underlying the findings in their manuscript fully available?

Reviewer #3: Yes

5. Is the manuscript presented in an intelligible fashion and written in standard English?

Reviewer #3: Yes

Reviewer #3: The authors have satisfactorily addressed the majority of your prior comments from both review rounds. The manuscript is now more coherent, methodologically sound, and appropriately cautious in interpretation. A few minor enhancements (deeper caution in discussing meta-regression results) could still be encouraged, but these do not warrant further major revision.

**Do you want your identity to be public for this peer review?** For information about this choice, including consent withdrawal, please see our Privacy Policy

Reviewer #3: No

---

## [Editor Report · Acceptance letter]

PONE-D-25-37457R2

PLOS One

Dear Dr. Zarrinkamar,

I'm pleased to inform you that your manuscript has been deemed suitable for publication in PLOS One. Congratulations! Your manuscript is now being handed over to our production team.

Kind regards,

on behalf of

Dr. Ozra Tabatabaei-Malazy

Academic Editor

PLOS One